# Effects of a Low-Fat Vegan Diet on Gut Microbiota in Overweight Individuals and Relationships with Body Weight, Body Composition, and Insulin Sensitivity. A Randomized Clinical Trial

**DOI:** 10.3390/nu12102917

**Published:** 2020-09-24

**Authors:** Hana Kahleova, Emilie Rembert, Jihad Alwarith, Willy N. Yonas, Andrea Tura, Richard Holubkov, Melissa Agnello, Robynne Chutkan, Neal D. Barnard

**Affiliations:** 1Physicians Committee for Responsible Medicine, Washington, DC 20016, USA; erembert@pcrm.org (E.R.); jihad93@vt.edu (J.A.); willynatanael@gmail.com (W.N.Y.); nbarnard@pcrm.org (N.D.B.); 2Metabolic Unit, CNR Institute of Neuroscience, 35127 Padua, Italy; andrea.tura@cnr.it; 3School of Medicine, University of Utah, Salt Lake City, UT 84132, USA; richard.holubkov@hsc.utah.edu; 4uBiome Inc., San Francisco, CA 94103, USA; agnellom@gmail.com; 5Department of Gastroenterology, Georgetown MedStar Hospital, Washington, DC 20007, USA; rchutkan@gmail.com; 6Adjunct Faculty, George Washington University School of Medicine and Health Sciences, Washington, DC 20052, USA

**Keywords:** diet, gut microbiome, nutrition, vegan, weight loss, obesity

## Abstract

Diet modulates gut microbiota and plays an important role in human health. The aim of this study was to test the effect of a low-fat vegan diet on gut microbiota and its association with weight, body composition, and insulin resistance in overweight men and women. We enrolled 168 participants and randomly assigned them to a vegan (*n* = 84) or a control group (*n* = 84) for 16 weeks. Of these, 115 returned all gut microbiome samples. Gut microbiota composition was assessed using uBiome Explorer™ kits. Body composition was measured using dual energy X-ray absorptiometry. Insulin sensitivity was quantified with the predicted clamp-derived insulin sensitivity index from a standard meal test. Repeated measure ANOVA was used for statistical analysis. Body weight decreased in the vegan group (treatment effect −5.9 kg [95% CI, −7.0 to −4.9 kg]; *p* < 0.001), mainly due to a reduction in fat mass (−3.9 kg [95% CI, −4.6 to −3.1 kg]; *p* < 0.001) and in visceral fat (−240 cm^3^ [95% CI, −345 to −135 kg]; *p* < 0.001). PREDIcted M, insulin sensitivity index (PREDIM) increased in the vegan group (treatment effect +0.83 [95% CI, +0.48 to +1.2]; *p* < 0.001). The relative abundance of Faecalibacterium prausnitzii increased in the vegan group (+5.1% [95% CI, +2.4 to +7.9%]; *p* < 0.001) and correlated negatively with changes in weight (r = −0.24; *p* = 0.01), fat mass (r = −0.22; *p* = 0.02), and visceral fat (r = −0.20; *p* = 0.03). The relative abundance of Bacteroides fragilis decreased in both groups, but less in the vegan group, making the treatment effect positive (+18.9% [95% CI, +14.2 to +23.7%]; *p* < 0.001), which correlated negatively with changes in weight (r = −0.44; *p* < 0.001), fat mass (r = −0.43; *p* < 0.001), and visceral fat (r = −0.28; *p* = 0.003) and positively with PREDIM (r = 0.36; *p* < 0.001), so a smaller reduction in Bacteroides fragilis was associated with a greater loss of body weight, fat mass, visceral fat, and a greater increase in insulin sensitivity. A low-fat vegan diet induced significant changes in gut microbiota, which were related to changes in weight, body composition, and insulin sensitivity in overweight adults, suggesting a potential use in clinical practice.

## 1. Introduction

The gut microbiota play an important role in human physiology and health. An imbalance in the microbiome has been associated with various conditions, including obesity and type 2 diabetes [1]. Evidence from previous studies suggests that obese people have a reduced number of bacterial species (“bacterial species richness”) [2] and relatively less abundance of the phyla *Bacteroidetes* compared to their lean counterparts [3]. Additionally, a reduced ratio of *Bacteroidetes* to *Firmicutes* has been shown to be associated with obesity and other related metabolic disorders [4].

Diet composition is known to modulate gut microbiota composition [5,6]. For example, long-term adherence to a predominantly plant-based dietary pattern (e.g., a vegetarian or vegan diet) leads to altered gut microbiota composition, compared with that of omnivores [7]. A vegan diet is characterized by elimination of animal products and is based mainly on the consumption of grains, legumes, vegetables, and fruits. Vegetarian and vegan diets have been shown to be effective in weight management [8] and reduce the risk of developing metabolic syndrome and diabetes [9]. However, information on the link between a vegan diet and gut microbiota is limited and mostly based on observational studies [7]. Little information is available on how the interaction between diet and the gut may, in turn, be associated with metabolic effects.

In this randomized clinical trial, we explored the effects of a low-fat vegan diet on gut microbiota composition and, in turn, how changes in the microbiota were associated with changes in body weight, body composition, and insulin resistance in overweight individuals. This manuscript is associated with a larger trial [10] and tests the associations of the gut microbiome with changes in body weight, body composition, and insulin resistance. We hypothesized that the following four outcomes would occur after 16 weeks on a low-fat vegan diet: an increase in the abundance of *Bacteroidetes* relative to *Firmicutes* and increases in the relative abundances of *Bacteroides fragilis*, *Prevotella*, and *Faecalibacterium Prausnitzii*.

## 2. Materials and Methods

### 2.1. Participants and Methods

Overweight participants (*n* = 168, a subset of the original study population of *n* = 244) were randomly assigned to follow a low-fat vegan (*n* = 84) or a control diet (*n* = 84). Body composition was measured using Dual X-ray Absorptiometry. We calculated insulin sensitivity using the PREDIM index (predicted clamp-derived insulin sensitivity index from a standard meal test). Stool microbiome composition was assessed using the uBiome Explorer™ microbiome sequencing kits (uBiome, Inc., San Francisco, CA, USA).

### 2.2. Study Design

The study was conducted between July 2017 and May 2018 using a single-center, randomized, open parallel design. Overweight, but otherwise healthy adult men and women with a body-mass index between 28 and 40 kg/m^2^ were enrolled. They were recruited through local newspaper advertisements, radio advertisements, healthcare professional referrals, mailing lists, and flyers. Exclusion criteria included a history of diabetes, smoking, alcohol or drug abuse, pregnancy or lactation, and current adherence to a vegan diet. The study protocol was approved by the Chesapeake Institutional Review Board on 12 October 2016 and 11 May 2017 (Pro00018983). All participants provided written informed consent. Registration on ClinicalTrials.gov was initiated on 20 October 2016 (Identifier: NCT02939638). Because this aspect of the research was added after study onset, only a subset of 168 participants out of the overall study population was enrolled. This article follows the Consolidated Standards of Reporting Trials (CONSORT) reporting guideline.

### 2.3. Randomization and Study Groups

Participants were randomly assigned in a 1:1 ratio to a vegan or a control group based on a computer-generated randomization protocol. The randomization protocol was not to be accessible beforehand. The participants were not blinded to their group assignment. They were examined at baseline and 16 weeks. The vegan group was asked to follow a low-fat vegan diet consisting of vegetables, grains, legumes, and fruits. They were instructed to avoid animal products and added oils. Daily fat intake was limited to 20–30 g. The dietary instructions were given at weekly classes and the participants were encouraged to attend them. No meals were provided. Vitamin B_12_ was supplemented (500 µg/day). Participants in the control group were asked to maintain their current diets, which included animal products, for the duration of the study. Laboratory measurements, gut microbiome analyses, and statistical analyses were made by staff members who were blind to group assignment.

### 2.4. Dietary Intake and Physical Activity

Each participant completed a 3 day dietary record at baseline and again at 16 weeks. Dietary intake data were collected and analyzed by a certified study staff member using the Nutrition Data System for Research version 2016, developed by the Nutrition Coordinating Center (NCC), University of Minnesota, Minneapolis, MN [11]. Random 24 h dietary recalls were also performed by registered dietitians at weeks 3 and 8 to ascertain adherence. The study participants were instructed not to change their physical activity and to continue their usual medications, except as modified by their personal physicians. Physical activity was assessed by the International Physical Activity Questionnaire (IPAQ) [12].

### 2.5. Gut Microbiota Composition

uBiome Explorer^TM^ kits (uBiome, Inc., San Francisco, CA, USA) were distributed to participants at each assessment. Participants followed the instructions in the kit to collect a stool sample by transferring a small amount of stool into the collection tube using the included sterile swab. The collection tubes contained a lysis and stabilization buffer. Samples were then mailed to uBiome’s laboratory for sequencing and analysis. DNA extraction and next-generation sequencing of the V4 region of the 16S rRNA gene were performed at uBiome as described previously [13]. For DNA extraction, samples were first lysed via bead-beating and DNA was extracted in a class 1000 clean room using a liquid-handling robot by a guanidine thiocyanate silica column-based purification method (Hummel, Cady). The V4 variable region of the 16S rRNA gene was amplified via PCR with universal primers (515F: GTGCCAGCMGCCGCGGTAA and 806R: GGACTACHVGGGTWTCTAAT) [Caparoso] as well as Illumina barcodes. PCR products were pooled, column-purified, and size-selected through microfluidic DNA fractionation (Minalla). qPCR was used to quantify consolidated libraries using the Kapa Bio-Rad iCycler qPCR kit on a BioRad MyiQ before sequencing. Sequencing was performed in a pair-end modality on the Illumina NextSeq 500 platform rendering 2 × 150 bp paired-end sequences.

After sequencing, BCL2FASTQ software (version 1.8.4., Illumina, CA, USA) was used to demultiplex the samples and generate fastq files. Reads with Q-score < 30 were excluded from the analysis. After filtering, the primers were removed and forward and reverse reads were appended together and clustered using version 2.1.5 of the Swarm algorithm [14] with a distance of 1 nucleotide and the “fastidious” and “usearch-abundance” flags. The most abundant sequence per cluster was considered the real biological sequence and was assigned the count of all reads in the cluster. The remainder of the reads in a cluster were considered to contain errors as a product of sequencing. Chimera sequences were removed using the VSEARCH algorithm [15]. Reads passing all the above filters (filtered reads) were aligned using 100% identity over 100% of the length against a hand-curated database of target 16S rRNA gene sequences and taxonomic annotations derived from version 132 of the SILVA database [16,17]. The relative abundance of each taxon was determined by dividing the count linked to that taxa by the total number of filtered reads. Alpha diversity was calculated using an abundance-weighted phylogenetic diversity measure as described by McCoy et al. [18].

### 2.6. Anthropometric and Metabolic Measurements

All measurements were performed on an outpatient basis after a 10–12 h overnight water-only fast. Height was measured using a stadiometer. Weight was measured using a periodically calibrated digital scale accurate to 0.1 kg. Body composition was measured using a DXA scan (iDXA; GE Healthcare, Chicago, IL, USA).

Plasma concentrations of glucose, immunoreactive insulin, and C-peptide were measured postprandially at 0, 30, 60, 120, and 180 min after stimulation with a liquid breakfast (Boost Plus, Nestle, Vevey, Switzerland; 720 kcal, 34% of energy from fat, 16% protein, 50% carbohydrate). Serum glucose was analyzed using the Hexokinase UV endpoint method (Roche, Basel, Switzerland). Plasma immunoreactive insulin and C-peptide concentrations were determined using insulin and C-peptide electro-chemiluminescence immunoassay (ECLIA) kits (Roche, Basel, Switzerland). HbA1c was measured by turbidimetric inhibition immunoassay (Roche, Basel, Switzerland). Plasma lipids concentrations were measured by enzymatic colorimetric methods (Roche, Basel, Switzerland). PREDIM index, previously validated against clamp-derived measures of insulin sensitivity [19], was calculated as a measure of dynamic postprandial insulin sensitivity.

### 2.7. Statistical Analysis

Based on the power analysis, the trial required a minimum of 73 participants per arm for the primary outcome of the study, i.e., the thermic effect of food. For the current analysis with four primary gut microbiota outcomes and therefore, incorporating a Bonferroni correction for multiple comparisons, a two-sided test of significance (by *t*-test or two-way ANOVA) has 80% power to detect a significant treatment difference if the true treatment effect was of a magnitude of at least 0.56 of a standard deviation in the within-group changes over time for any of the four outcomes. For the more general exploratory analysis of all 50 outcomes examined, the study has 80% power to detect a treatment difference at the multiplicity-adjusted 0.001 significance level if the true treatment effect is at least 0.70 of a standard deviation. To detect a significant correlation between two factors in the exploratory analysis of relationships between gut microbiota and metabolic outcomes, using an alpha level Bonferroni-corrected for the 20 comparisons performed, the study has 80% power to detect a significant association in cases where the true correlation was of magnitude 0.31 or greater.

A repeated measure ANOVA model that included the factors group, subject, and time was used to test the between-group differences throughout the 16-week study. Interaction between group and time (Gxt) was calculated for each variable. We tested the data for normal distribution. Within each diet group, paired comparison *t*-tests were calculated to test whether the change from baseline to 16 weeks was significantly different from zero. The treatment effect size is the difference in outcomes, from baseline to week 16, between the vegan and control group. Bonferroni correction for multiple comparisons was used for four hypotheses at *p* = 0.0125 (0.05/4). The rest of the outcomes, performed in an exploratory manner, are presented for a complete overview. Pearson correlations were calculated for the relationship between changes in overall diversity and our four pre-defined characteristics of gut bacteria composition and changes in body weight, fat mass, visceral fat volume, and insulin sensitivity. Taking into account the 20 evaluations performed, only those associations with unadjusted *p*-values under *p* = 0.05/20 = 0.0025 were interpreted as rigorously significant.

## 3. Results

### 3.1. Characteristics of the Study Participants

Adults aged 25 to 75 years, with a body-mass index between 28 and 40 kg/m^2^, were enrolled. Out of 168 participants who were randomized, 151 (90%) completed the entire study, of whom 115 (68.5%) returned all their gut microbiome samples. The majority of our study participants (85%, *n* = 143) were women. The flow of participants through the study is shown in Figure 1 and their baseline characteristics are shown in Table 1. The attendance rate in weekly classes fluctuated between 68–93%. Changes in physical activity, dietary intake, anthropo-metabolic outcomes, and gut microbiome during the study are shown in Table 2. We present *p*-values from an unadjusted ANOVA model as well as using an ANOVA model adjusted for age and gender. Inclusion of information from subjects with incomplete data and/or an additional ANOVA model adjusted for physical activity and energy intake did not alter the results significantly.

### 3.2. Body Weight, Body Composition, and Insulin Sensitivity

Body weight fell significantly in the vegan group (treatment effect −5.9 kg [95% CI, −7.0 to −4.9kg]; *p* < 0.001), mainly due to a reduction in fat mass (treatment effect −3.9 kg [95% CI, −4.6 to −3.1 kg]; *p* < 0.001) and in visceral fat volume (treatment effect −240 cm^3^ [95% CI, −345 to −135 kg]; *p* < 0.001). PREDIM increased significantly (*p* < 0.001) in the vegan group (treatment effect +0.83 [95% CI, +0.48 to +1.2]; *p* < 0.001). Appendix A shows violin plots of the changes in metabolic outcomes by treatment arm.

### 3.3. Gut Microbiota Composition

Changes in gut microbiota composition are summarized in Table 2. The α-diversity, which is the measure of microbial diversity within each sample, remained unchanged in the vegan group and increased (*p* < 0.001) in the control group (treatment effect −0.20 [95% CI, −0.34 to −0.06]; *p* = 0.0043). The count of butyrate-producing bacteria did not change in the vegan group and was reduced (*p* = 0.03) in the control group, with no significant difference between the groups (treatment effect +8396 [95% CI, −4458 to +21249]; *p* = 0.20). The relative abundance of *Bacteroidetes* increased in the vegan group insignificantly (p = 0.06) and the difference between the groups was not statistically significant (treatment effect +2.8% [95% CI, −2.0 to +7.5%]; *p* = 0.25). The *Firmicutes* to *Bacteroidetes* ratio did not change significantly in either group. The relative abundance of *Faecalibacterium prausnitzii* increased in the vegan group (treatment effect +5.1% [95% CI, +2.4 to +7.9%]; *p* < 0.001). The relative abundance of *Bacteroides fragilis* decreased in both groups, but less in the vegan group, making the treatment effect positive (treatment effect +18.9% [95% CI, +14.2 to +23.7%]; *p* < 0.001).

### 3.4. The Relationship between Changes in Gut Microbiota and Metabolic Outcomes

Changes in microbial diversity correlated positively with changes in body weight (r = +0.19; *p* = 0.048) and negatively with changes in PREDIM (r = −0.23; *p* = 0.02). Relative changes in the abundance of *Faecalibacterium prausnitzii* correlated negatively with changes in body weight (r = −0.24; *p* = 0.01; Figure 2A), fat mass (r = −0.22; *p* = 0.02; Figure 2B), and visceral fat volume (r = −0.20; *p* = 0.03; Figure 2C). Relative changes in *Bacteroides fragilis* correlated negatively with changes in body weight (r = −0.44; *p* < 0.001; Figure 2D), fat mass (r = −0.43; *p* < 0.001; Figure 2E), and visceral fat volume (r = −0.28; *p* = 0.003; Figure 2F) and positively with changes in PREDIM (r = 0.36; *p* < 0.001). That is, a smaller reduction in relative abundance of *Bacteroides fragilis* was associated with a greater loss of body weight, fat mass, visceral fat, and a greater increase in insulin sensitivity.

In an additional analysis to assess whether baseline levels of the relative abundance of the key gut microbial outcomes were associated with a change in body weight, we noted a weak association with the *Bacteroidetes:Firmicutes* ratio (highest weight loss in the highest quartile; *p* = 0.03), a strong association with *Bacteroides fragilis* (weight loss decreasing with increasing quartiles; *p* < 0.001), and no association with *Faecalibacterium prausnitzii* and *Prevotella* (Appendix A).

## 4. Discussion

In this randomized trial, a low-fat vegan diet led to a significant reduction in body weight, primarily reflecting a reduction in fat mass and in visceral fat volume and a significant increase in insulin sensitivity. An increase in the relative abundance of *Faecalibacterium prausnitzii* in the vegan group and a smaller decrease in the relative abundance of *Bacteroides fragilis* in the vegan group, relative to the control group, correlated negatively with changes in body weight, fat mass, and visceral fat volume. That is, a smaller reduction in the relative abundance of *Bacteroides fragilis* was associated with a greater loss of body weight, fat mass, and visceral fat and a greater increase in insulin sensitivity. Additionally, changes in the relative abundance of *Bacteroides fragilis* correlated positively with changes in insulin sensitivity. The relative abundance of *Bacteroidetes* increased in the vegan group, but the difference between the groups was not statistically significant. The *Firmicutes* to *Bacteroidetes* ratio did not change significantly in either group.

### 4.1. Bacteroidetes and Diet

Our findings, particularly the increase in *Bacteroidetes* in response to a low-fat vegan diet, are in accordance with previous studies. A study including 11 vegetarians, 20 vegans, and 29 omnivores found higher abundance of *Bacteroidetes*, *Clostridium clostridioforme*, and *Faecalibacterium prausnitzii*, which are all considered to be beneficial, in the vegetarians and vegans compared with the omnivores [20]. Similarly, a study comparing the gut microbiome of 15 vegetarians (3 vegans and 12 lacto-ovo-vegetarians) with 14 non-vegetarians in Austria found higher abundance of *Bacteroidetes* in vegetarians and vegans [21].

Another study compared the bacterial composition between Indian and Chinese adults. While both populations followed diets based on grains, legumes, and vegetables, the Chinese diet was higher in animal fat and protein than the Indian diet of whole grains and plant-based foods. The percentage of *Bacteroidetes* within the microbiomes of Indian participants was nearly four times greater than in the Chinese, 16.39% versus 4.27%, respectively (*p* = 0.001), presumably due to their lower consumption of animal products [22].

### 4.2. Firmicutes to Bacteroidetes Ratio, Diet, and Body Weight

Decreased levels of *Firmicutes* in favor of *Bacteroidetes* and *Bifidobacteria* may be beneficial in preventing and treating obesity [23]. It has been demonstrated that the *Bacteroidetes* to *Firmicutes* ratio is negatively correlated with BMI: a low ratio, which is common in the context of a Western diet, is associated with a high BMI [24]. The *Bacteroidetes* phylum has been shown to be three fold less abundant in obese people compared to non-obese individuals. Additionally, *Firmicutes* abundance has been shown to be higher in obese people [25], although some studies have failed to confirm this relationship [26,27]. A possible explanation may be provided by the finding that a 20% increase in *Firmicutes* and a corresponding decrease in *Bacteroidetes* abundance are associated with a 150 kcal/day increase in energy harvest, resulting in weight gain over time [28]. Therefore, an increased *Bacteroidetes* to *Firmicutes* ratio, as seen with a high-fiber, plant-based diet, may result in weight loss by reducing the number of calories extracted from the diet [29]. In the present study, however, despite an increase in *Bacteroidetes* after 16 weeks of a low-fat vegan diet, the *Bacteroidetes* to *Firmicutes* ratio did not change, contrary to our hypothesis. However, we noted a weak association between weight loss and the baseline *Bacteroidetes* to *Firmicutes* ratio: the highest weight loss occurred in the highest quartile.

### 4.3. Prevotella, Diet, and Body Weight

Also in contrast to our hypothesis, we did not observe an increase in *Prevotella*, although both *Bacteroidetes* and *Prevotella* are characterized as polysaccharide-degrading bacteria [30]. A vegan diet provides a rich source of polysaccharides for bacterial substrate utilization [7]. Furthermore, the relative abundance of both *Bacteroidetes* and *Prevotella* has been shown to increase with vegetarian and vegan diets rich in dietary fiber, using PCR-based DNA profiling techniques [31]. A possible explanation is the low relative abundance of *Prevotella* compared with *Bacteroidetes* in our study population at baseline. It has been shown previously that the balance between the two may have more influence on body weight than either one alone [32].

### 4.4. Faecalibacterium, Diet, and Body Weight

The increase in relative abundance of *Faecalibacterium prausnitzii* in response to a low-fat vegan diet observed in our study is consistent with previous research on vegetarian diets [33]. We observed no association between weight loss and baseline relative abundance of *Faecalibacterium prausnitzii*, which suggests that the participants benefited metabolically from an increase in this bacteria, regardless of the baseline values. *Faecalibacterium prausnitzii* is more abundant in Native Africans, from the KwaZulu-Natal Province of South Africa, consuming more resistant starch, compared to African Americans who consume diets higher in protein and fat [34]. Likewise, resistant starch supplementation increases the abundance of bacteria closely related to *Faecalibacterium prausnitzii* and other species producing short-chain fatty acids [35], which degrade plant polysaccharides and starch to produce health-promoting butyrate [36].

In a 2015 study, both a high-fiber, vegetable-rich macrobiotic diet and a Mediterranean-style diet were associated with an increase in *Faecalibacterium* in patients with type 2 diabetes [36]. While these volunteers initially presented with lower baseline levels of this species than their non-diabetic counterparts, the 21-day dietary interventions resulted in an increase in *Faecalibacterium*. In addition, *Faecalibacterium* was inversely associated with fasting blood glucose in volunteers following the macrobiotic diet.

Other studies also report a lower abundance of *Faecalibacterium prausnitzii* in individuals with diabetes when compared to their non-diabetic counterparts [37]. This species was negatively associated with fasting blood glucose, HbA1c, and insulin resistance. *Faecalibacterium prausnitzii* is also negatively associated with inflammatory markers, such as C-reactive protein and interleukin−6 [37]. Similarly, Xu et al. reported a negative association between *Faecalibacterium prausnitzii* and fasting blood glucose, HbA1c, and postprandial blood glucose levels and a positive association with β-cell function [38]. These studies suggest a positive correlation between *Faecalibacterium prausnitzii* abundance and glucose control.

### 4.5. Bacteroides Fragilis, Diet, and Body Weight

Relative abundance of *Bacteroides fragilis* decreased in both groups, however less so in the vegan group. Our data show an association between a relative abundance of *Bacteroides fragilis* and changes in body weight, body composition, and insulin sensitivity. There was also a strong association between weight loss and baseline relative abundance of *Bacteroides fragilis*, i.e., a smaller magnitude of weight loss with increasing quartiles. Differences in relative abundance of *Bacteroides fragilis* have been shown in other studies. One study found higher abundance of *Bacteroides fragilis* with an omnivorous diet intervention compared to vegetarian and vegan diets [33]. Another study found a Japanese diet consisting of soybeans, radishes, cabbage, fish, seaweed, and green tea to result in lower *Bacteroides fragilis* counts when compared to a Western diet high in meat [39]. However, in contrast to our findings, an increase in *Bacteroides fragilis* after one month on a vegan diet was described previously in a small interventional trial in six obese participants with type 2 diabetes [28,37]. *Bacteroides fragilis* has been shown to utilize fiber extracts as a growth substrate [40]. Thus, the variations in its relative abundance may be due to a potential metabolic adaptation of *Bacteroides fragilis* to utilize carbohydrate and protein substrates for growth [41]. Furthermore, variations across the individual strains might explain these discrepancies in *Bacteroides fragilis* proliferation. Further research is needed to provide additional support to these findings and to examine systemic changes that may occur in response to changes in *Bacteroides fragilis* abundance.

### 4.6. Possible Mechanisms

The low-fat vegan diet used in our study may have influenced the gut microbiota through several possible mechanisms. A high-fiber diet increases the production of short-chain fatty acids in the gut, which have a positive effect on weight regulation and cardiometabolic health [42]. The low fat content of the vegan diet contributed to the reduction in energy intake, which may have influenced the gut microbiota [43]. Furthermore, the participants in the vegan group significantly changed the consumption of all macronutrients (carbohydrates, protein, and fat), all of which have been shown to independently influence the gut microbiome and have been reviewed in detail previously [44]. Finally, a vegan diet is rich in polyphenols and other phytochemicals, which can further modulate the composition of the gut microbiome [45].

### 4.7. Study Strengths and Limitations

The strengths of the study include the randomized parallel design, in which all participants started simultaneously, allowing the investigators to rule out possible effects of seasonal fluctuations in the diet. The study duration was reasonably long, providing sufficient time for adaptation to the diet and to capture microbiome changes. We used physiological stimulation by a standard mixed meal, enabling us to quantify insulin sensitivity during a physiological perturbation, with an index that has been validated against the hyperinsulinemic euglycemic clamp. The low attrition rate suggests that the intervention was acceptable and sustainable, in accordance with the findings of a previous long-term study [46]. Given that the participants were living at home and preparing their own meals or eating at restaurants, our results are applicable outside the research setting in free-living conditions.

The study also has important limitations. Dietary intake was calculated based on self-reported dietary records, which have well-known limitations [47]. However, it is reassuring that the reported changes in nutrient intake were paralleled by weight loss and metabolic changes. We studied the effects of a low-fat vegan diet and the vegan participants changed the consumption of all macronutrients at once. Due to the low-fat content, the energy intake decreased, which may have affected the gut microbiome by itself. Furthermore, we are not able to differentiate which macronutrients influenced the changes in gut microbiota. Our participants were generally health-conscious individuals who were willing to make substantial changes to their diet. In this regard, they may not be representative of the general population, but may be representative of a clinical population seeking help for weight problems. We observed significant changes in gut microbiota parameters within the control group participants. Those changes might have been the result of seasonal changes in the diet as well as the greater variability of the dietary intake in the control group compared with the vegan group, but they may also suggest that additional lifestyle factors may have influenced the composition of gut microbiota, apart from dietary intake. Gut microbiota composition may also be influenced by age. However, the most prominent changes happen during infancy and old age, not during adulthood [48]. Furthermore, our ANOVA model adjusted for age did not change the results in a significant way. Bonferroni correction for multiple comparisons was used for four hypotheses at *p* = 0.0125 (0.05/4). The rest of the outcomes, performed in an exploratory manner, are presented for complete overview. However, if all the changes in the presented variables had been considered formal outcomes in the study, an extremely conservative assessment would have treated only those with a significance level of *p* = 0.001 (0.05/50) as of true statistical significance. Lastly, some of the correlations were relatively weak and should be interpreted cautiously.

### 4.8. Practical Implications

Vegetarian and vegan diets have been shown to be effective in weight management and in diabetes prevention and treatment [8,49,50]. This study has explored the link between changes in the gut microbiome in response to a 16 week low fat vegan dietary intervention and changes in body weight, body composition, and insulin sensitivity. We have demonstrated that a low-fat vegan diet elicited changes in gut microbiome that were associated with weight loss, a reduction in fat mass and visceral fat volume, and an increase in insulin sensitivity.

## 5. Conclusions

A shift to a low-fat vegan diet led to an increased relative abundance of *Faecalibacterium prausnitzii* and a smaller decrease, compared to the control group, in the relative abundance of *Bacteroides fragilis*, both of which correlated negatively with changes in body weight, fat mass, and visceral fat volume. Additionally, changes in the relative abundance of *Bacteroides fragilis* correlated positively with changes in insulin sensitivity. Our results suggest that changes in body weight, body composition, and insulin sensitivity in overweight adults after a low-fat vegan diet are related to changes in gut microbiota composition.

## Figures and Tables

**Figure 1 nutrients-12-02917-f001:**
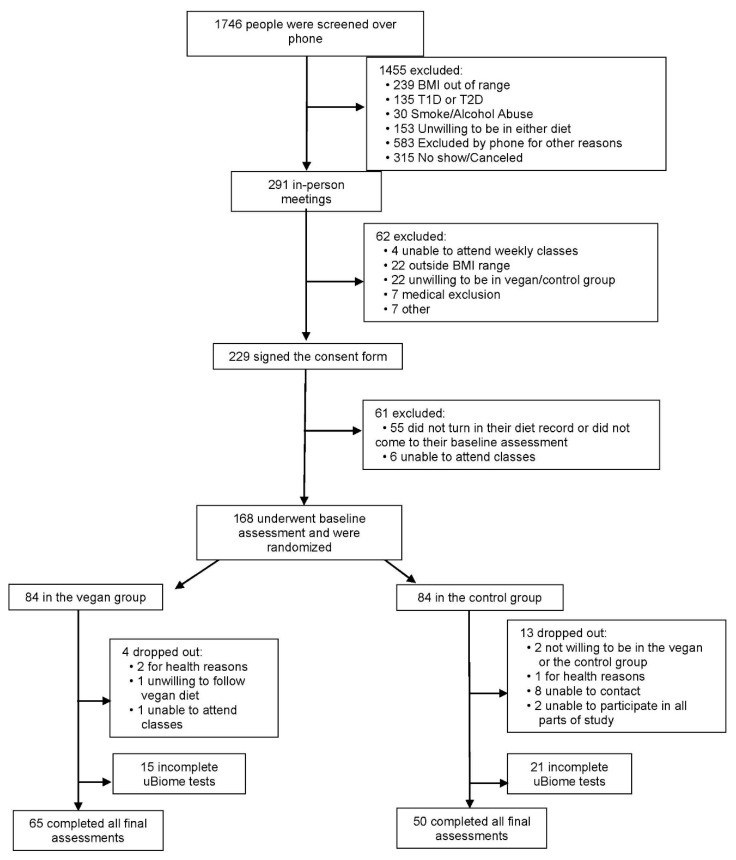
Enrollment of the Participants and Completion of the Study. T1D: type 1 diabetes; T2D: type 2 diabetes.

**Figure 2 nutrients-12-02917-f002:**
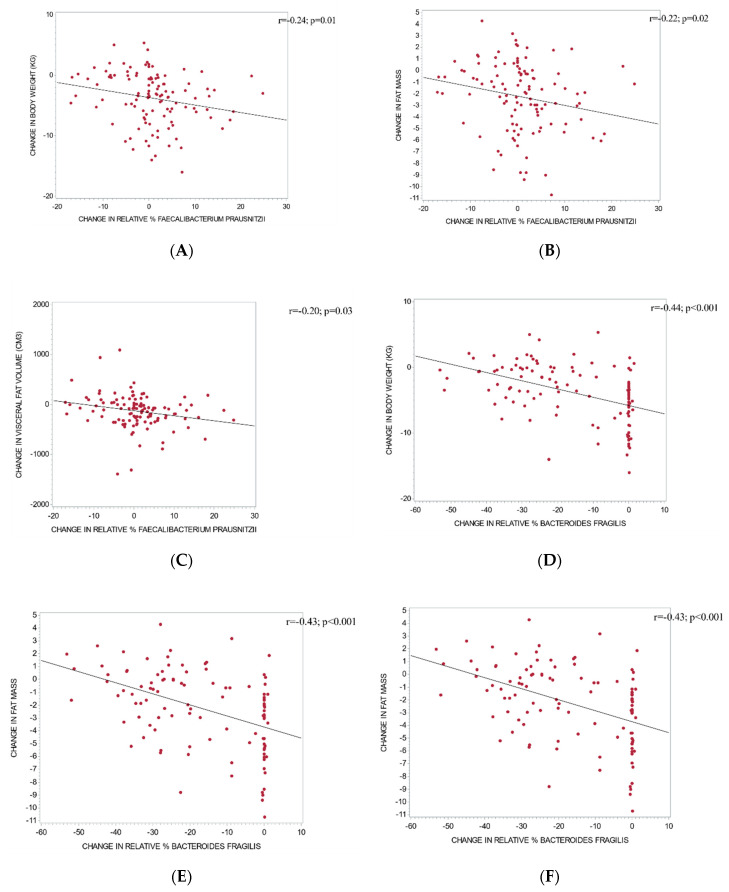
Correlations between changes in relative abundance of gut bacteria and changes in metabolic outcomes: Correlations between changes in relative abundance of *Faecalibacterium prausnitzii* and changes in body weight (**A**), fat mass (**B**), and visceral fat (**C**); and correlations between changes in the relative abundance of *Bacteroides fragilis* and changes in body weight (**D**), fat mass (**E**), and visceral fat (**F**).

**Table 1 nutrients-12-02917-t001:** Baseline characteristics of the Study Population.

Characteristics	Vegan Group (*n* = 84)	Control Group (*n* = 84)	Statistic
Age (years)	52.9 ± 11.7	57.5 ± 10.2	*p* = 0.01
Sex (number, %)			
Male	15 (17.9)	10 (11.9)	*p* = 0.28
Female	69 (82.1)	74 (88.1)	
BMI (kg/m^2^)	32.6 ± 3.7	33.6 ± 3.8	*p* = 0.10
Race, (number, %)			
White	42 (50.0)	41 (48.8)	*p* = 1.00
Black	40 (47.6)	39 (46.4)	
Asian, Pacific Islander	1 (1.2)	2 (2.4)	
Did not disclose	1 (1.2)	2 (2.4)	
Ethnicity, (number, %)			
Non-Hispanic	64 (76.2)	69 (82.1)	*p* = 0.74
Hispanic	5 (5.9)	4 (4.8)	
Did not disclose	15 (17.9)	11 (13.1)	
Marital status			
Not married	44 (52.4)	42 (50.0)	*p* = 0.27
Married	39 (46.4)	35 (41.7)	
Did not disclose	1 (1.2)	7 (8.3)	
Education			
High school	8 (9.5)	11 (13.1)	*p* = 0.20
Associates	0	1 (1.2)	
College	23 (27.4)	29 (34.5)	
Graduate degree	53 (63.1)	43 (51.2)	
Occupation			
Service occupation	22 (26.2)	14 (16.7)	*p* = 0.51
Technical, sales, administrative	19 (22.6)	22 (26.2)	
Professional or managerial	23 (27.4)	23 (27.4)	
Retired	12 (14.3)	18 (21.4)	
Other	8 (9.5)	7 (8.3)	
Medications (%)			
Lipid-lowering therapy	17 (20.2)	18 (21.4)	*p* = 0.85
Antihypertensive therapy	21 (25.0)	25 (29.8)	*p* = 0.49
Thyroid medications	10 (11.9)	7 (8.3)	*p* = 0.44

Data are means ± SD, or number (%). *p*-values refer to *t*-tests for continuous variables and χ^2^ for categorical variables. The *p*-value calculated for ethnicity distribution is for the comparison between Hispanic vs. non-Hispanic categories.

**Table 2 nutrients-12-02917-t002:** Changes in dietary intake, anthropo-metabolic outcomes, and gut microbiome during the study.

	Control Group		Vegan Group		Treatment Effect	*p* Value ^a^	*p* Value ^b^
	Baseline	Week 16	Baseline	Week 16			
Total Physical activity (metabolic equivalents)	2840 (2064–3616)	2107 (1555–2658) *	2982 (1696–4267)	2000 (1394–2606)	−248 (−1549 to +1053)	0.71	0.87
**Dietary Intake**							
Caloric Intake (kcal/day)	1726 (1606–1847)	1692 (1562–1821)	1827 (1689–1965)	1294 (1212–1376) ***	−498 (−696 to −300)	<0.001	<0.001
Total Fat (g/day)	72.2 (66.1–78.3)	71.5 (63.9–79.0)	74.1 (67.9–80.4)	24.3 (21.7–26.9) ***	−49.1 (−58.5 to −39.8)	<0.001	<0.001
Total Carbohydrate (g/day)	204 (187–221)	196 (178–214)	222 (202–241)	236 (219–254)	+22.6 (−6.5 to +51.7)	0.13	0.17
Total Protein (g/day)	67.2 (62.2–72.1)	68.9 (63.1–74.6)	69.0 (63.9–74.1)	42.9 (40.0–45.7) ***	−27.9 (−36.0 to −19.7)	<0.001	<0.001
Animal Protein (g/day)	37.5 (32.5–42.4)	39.3 (33.4–45.3)	38.5 (33.7–43.2)	1.2 (0.5–1.9) ***	−39.1 (−46.4 to −31.9)	<0.001	<0.001
Veg Protein (g/day)	29.7 (26.8–32.6)	29.5 (26.0–33.0)	30.6 (27.5–33.7)	41.7 (39.0–44.3) ***	+11.3 (+6.6 to +16.0)	<0.001	<0.001
Cholesterol (mg/day)	227 (190–263)	243 (201–285)	227 (196–258)	5.1 (3.3–6.9) ***	−238 (−285 to −191)	<0.001	<0.001
Total SFA (g/day)	21.7 (19.3–24.0)	21.9 (18.9–24.9)	23.3 (20.4–26.1)	4.8 (4.2–5.4) ***	−18.7 (−22.7 to −14.8)	<0.001	<0.001
Total MUFA (g/day)	26.7 (24.1–29.3)	25.8 (23.0–28.5)	26.4 (24.2–28.6)	7.9 (7.0–8.7) ***	−17.6 (−21.2 to −14.0)	<0.001	<0.001
Total PUFA (g/day)	17.9 (16.4–19.5)	17.8 (15.8–19.9)	18.4 (16.7–20.2)	9.0 (7.9–10.2) ***	−9.3 (−12.1 to −6.6)	<0.001	<0.001
Total Dietary Fiber (g/day)	23.1 (21.0–25.2)	23.1 (20.7–25.4)	24.0 (21.6–26.5)	33.2 (30.8–35.6) ***	+9.2 (+5.6 to +12.8)	<0.001	<0.001
Soluble Fiber (g/day)	6.0 (5.5–6.5)	6.6 (6.0–7.2) *	6.9 (6.2–7.5)	8.5 (7.8–9.2) ***	+1.0 (+0.1 to +2.0)	0.03	0.01
Insoluble Fiber (g/day)	17.0 (15.3–18.8)	16.4 (14.5–18.2)	17.1 (15.1–19.0)	24.6 (22.7–26.4) ***	+8.2 (+5.2 to +11.1)	<0.001	<0.001
**Anthropo-Metabolic Outcomes**							
Weight (kg)	93.4 (90.1–96.7)	92.9 (89.6–96.3)	92.9 (89.7–96.1)	86.5 (83.5–89.5) ***	−5.9 (−7.0 to −4.9)	<0.001	<0.001
BMI (kg/m^2^)	33.6 (32.6–34.5)	33.4 (32.4–34.4)	32.6 (31.8–33.5)	30.5 (29.6–31.3) ***	−2.0 (−2.4 to −1.6)	<0.001	<0.001
Fat Mass (kg)	42.0 (39.7–44.2)	41.7 (39.4–44.1)	39.8 (37.7–42.0)	35.7 (33.6–37.9) ***	−3.9 (−4.6 to −3.1)	<0.001	<0.001
VAT Volume (cm^3^)	1590 (1365–1814)	1589 (1360–1818)	1511 (1291–1732)	1271 (1084–1457) ***	−240 (−345 to −135)	<0.001	<0.001
PREDIM (mg/min/kg)	4.4 (4.0–4.8)	4.2 (3.8–4.6)	4.0 (3.7–4.3)	4.8 (4.4–5.1) ***	+0.83 (+0.48 to +1.2)	<0.001	<0.001
**Gut Microbiota Composition**							
Diversity	1.6 (1.5–1.7)	1.8 (1.7–1.9) ***	1.7 (1.6–1.8)	1.7 (1.6–1.8)	−0.20 (−0.34 to −0.06)	0.0043	0.003
Firmicutes	60,997 (41,121–80,874)	60,599 (51,612–69,586)	52,038 (41,221–62,856)	64,734 (53,368–76,100)	+13,094 (−11,808 to +37,996)	0.30	0.42
Firmicutes %	52.2 (48.7–55.7)	49.8 (46.6–52.9)	55.4 (51.7–59.1)	54.6 (51.4–57.9)	+1.7 (−2.7 to +6.1)	0.45	0.72
Bacteroidetes	41,690 (33,401–49,980)	53,871 (41,691–66,051)	31,944 (24,747–39,141)	42,855 (34,088–51,622) *	−1270 (−17,950 to +15,409)	0.88	1.00
Bacteroidetes %	37.6 (34.1–41.1)	38.3 (34.9–41.7)	31.6 (28.4–34.8)	35.1 (31.4–38.8)	+2.8 (−2.0 to +7.5)	0.25	0.06
Enterobacteriaceae	600 (−109–1308)	709 (−153–1570)	293 (135–451)	683 (−79.0–1444)	+280.6 (−545 to +1106)	0.50	0.44
Enterobacteriaceae %	0.47 (0.02–0.93)	0.66 (0.02–1.3)	0.73 (0.03–1.4)	0.47 (0.02–0.92)	−0.44 (−1.3 to +0.45)	0.33	0.25
Firmicutes:Bacteroidetes ratio	3.0 (0.38–5.6)	2.0 (0.92–3.1)	2.4 (1.6–3.1)	2.3 (1.7–2.9)	+0.90 (−0.76 to +2.6)	0.28	0.26
Butyrate producing bacteria	41,781 (34,410–49,152)	34358 (27,310–41,407) *	37,169 (29,540–44,799)	38,143 (30,371–45,915)	+8396 (−4458 to +21249)	0.20	0.32
Butyrate producing bacteria %	22.0 (19.5–24.5)	19.4 (16.8–21.9)	23.4 (21.5–25.2)	21.2 (19.2–23.2)	+0.48 (−4.0 to +5.0)	0.83	0.74
Prevotella	240 (104–376)	554 (49.4–1059)	224 (111–336)	402 (169–636)	−135 (−711 to +439)	0.64	0.61
Prevotella %	0.35 (0.09–0.60)	1.15 (0.08–2.2)	0.69 (0.01–1.4)	0.84 (0.02–1.7)	−0.65 (−2.1 to +0.8)	0.39	0.56
Akkermansia	1089 (495–1684)	5073 (66.0–10080)	2256 (640–3871)	3215 (1873–4557)	−3024 (−8211 to +2163)	0.25	0.26
Akkermansia %	1.4 (0.74–2.1)	2.4 (0.85–4.0)	2.1 (1.0–3.1)	2.3 (1.5–3.2)	−0.74 (−2.6 to +1.1)	0.43	0.40
Faecalibacterium prausnitzii	6935 (4905–8966)	7142 (4502–9783)	6304 (4127–8481)	12405 (8417–16394) *	+5895 (+506 to +11283)	0.03	0.17
Faecalibacterium prausnitzii %	7.2 (5.2–9.2)	5.3 (3.8–6.9)	5.5 (4.3–6.8)	8.8 (7.0–10.6) ***	+5.1 (+2.4 to +7.9)	0.0003	0.002
Bacteroides fragilis	31212 (23918–38505)	602 (−275–1478) ***	10641 (5513–15769)	524 (−19.0–1067) ***	+20493 (+11790 to +29195)	<.001	<.001
Bacteroides fragilis %	27.1 (23.6–30.6)	0.3 (0.0–0.60) ***	8.3 (5.1–11.5)	0.40 (0.10–0.70) ***	+18.9 (+14.2 to +23.7)	<.001	<.001
Clostridium	844 (529–1160)	956 (705–1206)	631 (432–829)	880 (694–1066)	+138 (−245 to +521)	0.48	0.62
Clostridium %	0.72 (0.54–0.90)	0.74 (0.58–0.90)	0.68 (0.55–0.81)	0.76 (0.63–0.88)	+0.05 (−0.17 to +0.27)	0.63	0.71
Methanobrevibacter	71.4 (22.9–120)	506 (156–856) *	299 (37.1–560)	826 (−2.8–1655)	+93.1 (−817 to +1003)	0.84	0.51
Methanobrevibacter %	0.09 (0.03–0.14)	0.35 (0.14–0.57) **	0.23 (0.07–0.39)	0.57 (0.18–0.96)	+0.07 (−0.35 to +0.49)	0.74	0.25
Eubacterium	0.42 (0.05–0.79)	2.0 (0.37–3.7) *	1.0 (0.08–2.0)	0.9 (0.03–1.8)	−1.7 (−3.5 to +0.09)	0.06	0.05
Eubacterium %	0.001 (0.00002–0.001)	0.002 (0.0002–0.003)	0.0009 (0.0001–0.002)	0.002 (−0.0005–0.004)	−0.0003 (−0.002 to +0.002)	0.77	0.88
Bifidobacterium	1254 (412–2096)	1712 (764–2659)	1278 (738–1818)	2313 (1286–3339) *	+577 (−700 to +1854)	0.37	0.89
Bifidobacterium %	1.4 (0.48–2.3)	1.8 (0.35–3.3)	1.3 (0.79–1.8)	1.7 (1.1–2.3)	−0.05 (−0.91 to +0.81)	0.91	0.64
Proteobacteria	5278 (3672–6883)	5195 (3305–7086)	3561 (2503–4619)	4165 (2798–5533)	+686 (−1622 to +2994)	0.56	0.78
Proteobacteria %	4.9 (3.7–6.0)	4.0 (3.0–5.1)	4.4 (3.2–5.5)	3.1 (2.4–3.9) *	−0.40 (−2.0 to +1.2)	0.61	0.58
Actinobacteria	2736 (1395–4077)	2247 (1261–3233)	2459 (1716–3203)	2729 (1658–3800)	+758 (−1071 to +2588)	0.41	0.85
Actinobacteria %	3.1 (1.9–4.4)	2.9 (1.2–4.7)	2.6 (2.0–3.1)	2.4 (1.6–3.2)	+0.05 (−1.4 to +1.5)	0.94	0.91
Ruminococcaceae	15,842 (11,727–19,956)	15775 (12,261–19,289)	18,807 (14,337–23,277)	22,265 (17,204–27,326)	+3525 (−4962 to +12011)	0.41	0.96
Ruminococcaceae %	15.4 (13.0–17.8)	12.9 (10.5–15.2)	19.0 (16.7–21.3)	18.5 (16.2–20.8)	+2.1 (−1.4 to +5.5)	0.24	0.66
Lachnospiraceae	26,182 (20,879–31,485)	18,224 (12,241–24,207) **	22,871 (18,631–27,111)	15,881 (11,212–20,550) *	+968 (−7106 to +9042)	0.81	0.79
Lachnospiraceae %	24.4 (22.1–26.7)	15.4 (11.3–19.4) ***	24.8 (22.5–27.1)	16.9 (13.6–20.2) ***	+1.2 (−4.6 to +7.0)	0.69	0.91
Roseburia	5993 (4634–7351)	5639 (4367–6912)	5821 (4255–7387)	6649 (4953–8345)	+1181 (−1469 to +3831)	0.38	0.34
Roseburia %	5.8 (4.7–6.9)	5.1 (4.1–6.1)	6.4 (5.3–7.5)	6.0 (4.8–7.2)	+0.28 (−1.6 to +2.2)	0.77	0.90
Anaerostipes	1502 (1020–1985)	2061 (1496–2625)	1481 (1103–1858)	2203 (1614–2792) *	+164 (−772 to +1100)	0.73	0.45
Anaerostipes %	1.4 (1.1–1.8)	1.5 (1.2–1.8)	1.6 (1.2–1.9)	1.8 (1.4–2.1)	+0.10 (−0.48 to +0.69)	0.72	0.33
Megasphaera	324 (−264–912)	334 (−270–938)	60.7 (19.5–102)	123 (−9.2–255)	+51.8 (−58.0 to +162)	0.35	0.80
Megasphaera %	0.72 (−0.67–2.1)	0.62 (−0.42–1.7)	0.12 (0.00–0.25)	0.19 (−0.05–0.42)	+0.16 (−0.24 to +0.56)	0.42	0.67

Data are means with 95% confidence interval (CI) mean. The treatment effect, which is tested by the group-by-time interaction in a repeated measures ANOVA model, does effectively compare the differences in outcomes from baseline to week 16 between the two study arms. Bonferroni correction for multiple comparisons was used for four hypotheses at *p* = 0.0125 (0.05/4). The rest of the outcomes, performed in an exploratory manner, are presented for complete overview. The treatment effect is the difference in outcomes, from baseline to week 16, between the vegan and control group. Diversity is defined as alpha diversity, which was calculated using an abundance-weighted phylogenetic diversity measure. Values listed under gut microbiota composition values are read counts and relative abundances (%). Relative abundance was calculated by dividing the number of reads assigned to each taxon by the total number of filtered reads. Listed *p* values are for interactions assessed by repeated measures ANOVA, first unadjusted (*p*-value ^a^: unadjusted, group x time) and then adjusted for age and gender (*p*-value ^b^, including all study participants). * *p* < 0.05, ** *p* < 0.01, and *** *p* < 0.001 for within-group changes from baseline assessed by paired comparison t tests. Abbreviations: SFA, saturated fatty acids; MUFA, monounsaturated fatty acids; PUFA, polyunsaturated fatty acids; BMI, body mass index; VAT, visceral adipose tissue; PREDIM (PREDIcted M, insulin sensitivity index).

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
