# Peer review of "Effects of a Low-Fat Vegan Diet on Gut Microbiota in Overweight Individuals and Relationships with Body Weight, Body Composition, and Insulin Sensitivity. A Randomized Clinical Trial"

_nutrients, 2020, doi:10.3390/nu12102917_

Round 1

Reviewer 1 Report

It is necessary to explain the contents of the low-fat plant-based diet and to examine the relationship between each ingredient and changes in intestinal microbiota. It is very important for the reader to show what changed the gut microbiota.

Also, the abstract is too long. The images in Fig. 1 and 2 are rough, so please increase the resolution.

Author Response

Thank you for your insightful comments and for your wonderful help in improving our manuscript! Please find our response attached.

Reviewer 1

  1. It is necessary to explain the contents of the low-fat plant-based diet and to examine the relationship between each ingredient and changes in intestinal microbiota. It is very important for the reader to show what changed the gut microbiota.

Thank you for this excellent comment. We have added a section on possible mechanisms and have discussed the possible role of different diet components (lines 389-398).

  1. Also, the abstract is too long.

Thank you for pointing this out. We shortened the abstract from 386 to 316 words.

  1. The images in Fig. 1 and 2 are rough, so please increase the resolution.

Thank you. We have provided high-resolution images for Fig. 1 and 2.

Reviewer 2 Report

Authors conducted an interesting and elegant study showing that a plant-based dietary pattern (vegetarian or vegan diet), other than reducing body weight, BMI, fat mass and visceral fat and improve insulin sensivity, was also associated with changes in gut microbiota, particularly with an increase of Faecalibacterium prausnitzii and with a smaller decrease of Bacteroides fragilis in the plant-diet group.

Author Response

Thank you for your insightful comments and for your wonderful help in improving our manuscript! 

Reviewer 2

Authors conducted an interesting and elegant study showing that a plant-based dietary pattern (vegetarian or vegan diet), other than reducing body weight, BMI, fat mass and visceral fat and improve insulin sensivity, was also associated with changes in gut microbiota, particularly with an increase of Faecalibacterium prausnitzii and with a smaller decrease of Bacteroides fragilis in the plant-diet group.

Thank you for reviewing our manuscript and for your kind words of appreciation!

Reviewer 3 Report

This study is interesting and much needed in the area of dietary intake and the gut microflora. Due to the challenges inherit to performing dietary intervention studies, there is a lack of research in this area.

Overall comments:

Some areas of the manuscript must be further developed:

  • The authors talk about a plant-based dietary pattern and provide examples of a vegetarian diet or a vegan diet (line 56), but then, they define a plant-based vegan diet in the next sentence. The rest of the manuscript uses the term ‘plant-based diet’. Plant-based diets are diets that include higher amounts of plant-based foods, but they can contain animal products. Using the term plant-based vegan diet is redundant, given that the basis of a vegan diet is plant-based foods. Vegan diet should be used if there is no inclusion of animal products and the authors should make it clear what the dietary intervention used in this study was based on. Was it a vegan diet or a plant-based diet.
  • Although the authors describe a reduced ratio of Bacteroidetes to Firmicutes as being associated with obesity, there is no discussion of several other studies that have not found any associations. For example:
    • Schwiertz A, Taras D, Schafer K, Beijer S, Bos NA, Donus C, et al. Microbiota and SCFA in lean and overweight healthy subjects. Obesity (Silver Spring) 2010;18:190–195. doi: 10.1038/oby.2009.167.
    • Duncan SH, Lobley GE, Holtrop G, Ince J, Johnstone AM, Louis P, et al. Human colonic microbiota associated with diet, obesity and weight loss. Int J Obes. 2008;32:1720–1724. doi: 10.1038/ijo.2008.155.
    • Correlation between body mass index and gut concentrations of Lactobacillus reuteri, Bifidobacterium animalis, Methanobrevibacter smithiiand Escherichia coli. Int J Obes. 2013;37:1460–1466. doi: 10.1038/ijo.2013.20.
  • The section ‘Randomization and Study groups’ does not adequately describe the procedures associated with the vegan group. Figure 1 mentions weekly classes, but that is not described in the methodology. Were participants in the vegan group required to attend weekly classes? What was the attendance rate? Were there any recommendations made related to physical activity? What was the purpose of assessing physical activity?
  • What was the age range of the participants? This information should be reported in the results and briefly addressed in the discussion, given that age seems to be associated with changes in the gut microbiome.
  • Physical activity was assessed by IPAQ but it is not included in the results. Please include the physical activity data in table 2.
  • There were several dietary variables that changed in the vegan group during the study. More notably dietary fat intake, animal protein intake, and total fiber intake. The discussion of the article was focused on plant-based diets, but an important aspect of the dietary intervention was a reduction in calories which was achieved mainly by a reduction in total fat intake. Based on all dietary changes observed, it is not possible to conclude that a plant-based diet is the cause of the changes in gut microbiome variables. It could be that the drastic reduction of fat was the driver for the changes. This should be addressed in the discussion and limitations of the study.

Author Response

Thank you for your insightful comments and for your wonderful help in improving our manuscript! Please find our response attached.

Reviewer 3

This study is interesting and much needed in the area of dietary intake and the gut microflora. Due to the challenges inherit to performing dietary intervention studies, there is a lack of research in this area.

Overall comments:

Some areas of the manuscript must be further developed:

  1. The authors talk about a plant-based dietary pattern and provide examples of a vegetarian diet or a vegan diet (line 56), but then, they define a plant-based vegan diet in the next sentence. The rest of the manuscript uses the term ‘plant-based diet’. Plant-based diets are diets that include higher amounts of plant-based foods, but they can contain animal products. Using the term plant-based vegan diet is redundant, given that the basis of a vegan diet is plant-based foods. Vegan diet should be used if there is no inclusion of animal products and the authors should make it clear what the dietary intervention used in this study was based on. Was it a vegan diet or a plant-based diet.

Thank you for this excellent comment. We have clarified the terms throughout the manuscript and have used the term “low-fat vegan diet” to describe our dietary intervention.

  1. Although the authors describe a reduced ratio of Bacteroidetes to Firmicutes as being associated with obesity, there is no discussion of several other studies that have not found any associations. For example:

Schwiertz A, Taras D, Schafer K, Beijer S, Bos NA, Donus C, et al. Microbiota and SCFA in lean and overweight healthy subjects. Obesity (Silver Spring) 2010;18:190–195. doi: 10.1038/oby.2009.167.

Duncan SH, Lobley GE, Holtrop G, Ince J, Johnstone AM, Louis P, et al. Human colonic microbiota associated with diet, obesity and weight loss. Int J Obes. 2008;32:1720–1724. doi: 10.1038/ijo.2008.155.

Correlation between body mass index and gut concentrations of Lactobacillus reuteri, Bifidobacterium animalis, Methanobrevibacter smithiiand Escherichia coli. Int J Obes. 2013;37:1460–1466. doi: 10.1038/ijo.2013.20.

Thank you for this important comment. We have added a clarifying statement in the discussion, lines 326-327).

  1. The section ‘Randomization and Study groups’ does not adequately describe the procedures associated with the vegan group. Figure 1 mentions weekly classes, but that is not described in the methodology. Were participants in the vegan group required to attend weekly classes? What was the attendance rate? Were there any recommendations made related to physical activity? What was the purpose of assessing physical activity?

Thank you. We have clarified that the vegan group was receiving the dietary instruction in weekly classes (lines 93-94). The attendance rate has been added to the results (lines 182-183). All participants were asked not to change their physical activity (lines 103-104) and this was assessed in order to avoid a potential confounder.

  1. What was the age range of the participants? This information should be reported in the results and briefly addressed in the discussion, given that age seems to be associated with changes in the gut microbiome.

Thank you for raising this important point. We have added the age range in the results (line 178) and in the discussion (lines 422-425).

  1. Physical activity was assessed by IPAQ but it is not included in the results. Please include the physical activity data in table 2.

Thank you. Physical activity data have been added to Table 2, first line.

  1. There were several dietary variables that changed in the vegan group during the study. More notably dietary fat intake, animal protein intake, and total fiber intake. The discussion of the article was focused on plant-based diets, but an important aspect of the dietary intervention was a reduction in calories which was achieved mainly by a reduction in total fat intake. Based on all dietary changes observed, it is not possible to conclude that a plant-based diet is the cause of the changes in gut microbiome variables. It could be that the drastic reduction of fat was the driver for the changes. This should be addressed in the discussion and limitations of the study.

Thank you for this excellent comment. We have added a section on possible mechanisms in the discussion (lines 389-398). We have also expanded the study limitations (lines 411-415).

Round 2

Reviewer 3 Report

I believe that the authors adequately addressed some of the major concerns pointed out by this reviewer. 

Author Response

Thank you for your kind response and for all your help with our manuscript!